

# Iron deficiency anemia among children aged 2–5 years in southern Ethiopia: a community-based cross-sectional study

Alemselam Zebdewos Orsango[1,2], Wossene Habtu[3], Tadesse Lejisa[3], Eskindir Loha[1,2,4], Bernt Lindtjørn[1,2] and Ingunn Marie S. Engebretsen[2]

[1] School of Public Health, College of Medicine and Health Sciences, Hawassa University, Hawassa, Ethiopia
[2] Centre for International Health, University of Bergen, Bergen, Norway
[3] Ethiopian Public Health Institute, Addis Ababa University, Addis Ababa, Ethiopia
[4] Chr. Michelsen Institute, Bergen, Norway

Corresponding author
Alemselam Zebdewos Orsango,
zalemselam@yahoo.com

## ABSTRACT

**Background:** Iron-deficiency anemia (IDA) is a common type of nutritional anemia in low-income countries, including Ethiopia. However, there is limited data on iron deficiency anemia prevalence and associated factors in Ethiopia, particularly for children aged 2 to 5 years.

**Objectives:** To establish the prevalence of iron deficiency anemia and associated risk factors, focusing on iron-rich food consumption among children aged 2 to 5 years in southern Ethiopia.

**Methods:** A community-based cross-sectional study was conducted in southern Ethiopia in 2017, involving 331 randomly selected children aged 2 to 5 years old. A structured questionnaire was used to collect information about the children and the households. Venous blood was collected from each child in a test tube to measure hemoglobin, ferritin, and C-reactive protein (CRP). Hemoglobin levels were determined using Hemocue®301 and adjusted for altitude. Anemia was defined as hemoglobin levels <11 g/dl. Ferritin was adjusted for inflammation based on CRP concentration and low ferritin concentration defined as adjusted ferritin concentration <12 µg/L. IDA was considered when a child had both hemoglobin level <11g/dl and low ferritin concentration. Bi-variable and multivariable logistic regression models were performed to identify factors associated with IDA and iron-rich food consumption.

**Results:** The prevalence of iron deficiency anemia was 25%, and the total anemia prevalence was 32%. Only 15% of children consumed iron-rich foods in the preceding 24 h, and 30% of children consumed iron-rich foods at least once in the preceding week. IDA decreased as the height for age z-score increased (Adjusted Odds Ratio 0.7; 95% CI [0.5–0.9]). Mothers with increased educational level (AOR 1.1; 1.0–1.2) and households with increased dietary diversity (AOR 1.4; 1.2–1.6) consumed more iron-rich foods.

**Conclusions:** Iron deficiency anaemia was a moderate public health problem in southern Ethiopia, and the iron-rich food consumption was low. Interventions should focus on food supplementation and fortification, food diversification and nutritional education, and promoting women's education.

## BACKGROUND

One-fourth of the global population is affected by anemia, and the highest prevalence is in preschool age children (*World Health Organization, 2015*). In Ethiopia, 56% of the children under the age of 5 years were anemic (*Central Statistical Agency, 2016*). It has long been assumed that up to half of the cases of anemia are due to iron deficiency (*World Health Organization, 2005*). But the recent review found that iron deficiency accounts for 25 percent of anemia in young children and 37 percent of anemia in women of reproductive age. Moreover, significant variations exist between countries, which may render generalized assumptions misleading (*Green et al., 2017*).

Children are particularly vulnerable to iron-deficiency anemia because of their increased iron requirements during periods of rapid growth, especially in the first five years of life. In children, iron deficiency can affect cognitive and motor development and increase susceptibility to infections (*Batra & Sood, 2005*).

In developing countries diets are dominated by starchy cereals that are low in lipids, proteins, vitamins, and minerals including iron (*Harika et al., 2017*). Poor families, which constitute the majority of the population in Ethiopia as well as in other low-income countries, often cannot afford animal food products that have high amounts of easily absorbable iron (*Gebreegziabiher, Etana & Niggusie, 2014*; *Kejo et al., 2018*). In Ethiopia, iron-rich food consumption by children is less than 21% (*Tiruneh et al., 2020*; *Herrador et al., 2015*) and more than 70% of the children scored low on dietary diversity assessments (*Temesgen et al., 2018*; *Worku et al., 2020*).

The current public health policy is supporting efforts to reduce child malnutrition in general and child anemia in particular, by promoting dietary diversity, deworming, iron supplementation during pregnancy, and malaria prevention (*Federal Democratic Republic of Ethiopia, 2018*). Despite these efforts, the prevalence of anemia among children less than five years old increased from 44% in 2011 to 57% in 2016 (*Central Statistical Agency, 2016*).

Diagnosing iron deficiency anemia is challenging in the presence of acute or chronic inflammations (*Grant et al., 2012*). In Ethiopia studies of IDA are often restricted by a shortage of laboratory facilities, and because of challenges in interpretation of biomarkers for IDA due to high burden of infections (*Namaste et al., 2017b*). We assumed there would be a high prevalence of IDA among children in the study area due to the high burden of anemia, limited access to iron supplementation and fortified foods, and lack of iron in the foods (*Central Statistical Agency, 2016*). In Ethiopia, literature on IDA is not only limited but also inconsistent. Different studies reported varying prevalence rates. The Ethiopia National Micronutrient Survey in 2014 estimated 8.6% of IDA prevalence at the national level among children 6–59 months old (*Ministry of Health, 2016*), and a study from southwest Ethiopia reported 37% of IDA among school children (*Desalegn, Mossie & Gedefaw, 2014*). Hence, determining the prevalence of iron deficiency anemia is

important to plan appropriate interventions and thereby to minimize future health risks (*Garcia-Casal et al., 2018*). This study aimed to establish the prevalence of iron- deficiency anemia and associated risk factors, focusing on the assessment of iron-rich food consumption among children aged 2 to 5 years in southern Ethiopia.

## METHODS

### Ethical consideration

The institution's ethical board of Hawassa University (IRB/098/08) and the Regional Ethical Committee West Norway (No. 2016/2034) provided ethical approvals. Hawassa University College of Medicine and Health Sciences approved to undertake field work (No.1471/09). Local administrative and health authorities of Hawassa City Administration Hawella Tula sub city Health Office (No. 3214/09) also granted official permission. Informed written consent was obtained from mothers for all study participants.

### Study area

This cross sectional study is a part of a larger study that aimed to evaluate the efficacy of home processed amaranth grain containing bread in the treatment of anemia among 2–5 year-old children in southern Ethiopia (*Orsango et al., 2020*). Thus, the Cheffe Cote Jebessa Kebele was purposively selected because amaranth grain grows as a wild crop in the village. The plant amaranth was used to prepare an alternative plant-based iron-rich food that is easily accessible and acceptable for children (*Orsango et al., 2020*; *Zebdewos et al., 2015*). The study area is located in a semi-urban part of Hawassa city in southern Nations Nationalities Peoples Regional state, 273 km south of Addis Ababa, the capital city of Ethiopia. The study was conducted from February 15, to March 30, 2017. According to a census we conducted before the start of our study, the population of the village was 23,010 people with 3,900 households, and 1,689 children aged 2–5 years old. Since it is a semi-urban area, some of the household members are farmers and some of them are engaged in non-agricultural activities in Hawassa city. The farmers cultivate maize, haricot beans, *ensete*, and Irish potatoes.

### Sampling procedure and sample size

The sample size of 340 children was calculated using OpenEpi software version 3.03 (*Sullivan, Dean & Soe, 2009*) considering the 33% national average prevalence of anemia among children aged 3–5 years old (*Central Statistical Authority, 2012*), 95% confidence level, and 5% margin of error. From the census list, 340 children were randomly selected using SPSS random generation technique (Fig. S1).

### Study participants

This cross-sectional study was done before the trial (*Orsango et al., 2020*), and we randomly selected 340 children aged 2 to 5 years and their mothers or caregivers, all of whom lived in the study area and who provided informed consent. We excluded children with conditions that precluded them from participating in measurements and venipuncture.

## Study variables

The primary outcome variable of this study was iron deficiency anemia, and secondary outcome variable was iron-rich food consumption. Independent variables of the study were socio-demography factors, economic status, dietary intake, and nutritional conditions of children.

## Blood collection and laboratory methods

Professional laboratory technicians have collected 3–5 ml of blood specimens from the vein by using a lithium heparin plasma separator test tube that used to measure hemoglobin, serum ferritin, and CRP (*Wei et al., 2010*). Hemoglobin was determined immediately on-site using HemoCue analyzers 301(Sweden). All samples with hemoglobin levels <11g/dl were tested further for serum ferritin and CRP concentration. For determination of serum ferritin level and CRP, the blood samples in the test tubes were immediately wrapped in aluminum foil, continuously shielded from light, and stored for 30 min at 4 °C until centrifugation. After centrifuging, serum was kept in plastic screw-capped tubes with the participant's identity and stored at −20 °C. Then the serum was transported in a cold box chain for analysis of ferritin and CRP to the Ethiopian Public Health Institute at Addis Ababa.

C-reactive protein (CRP) was measured on Cobas 6000 (c501 module) Roche (Germany) by applying enhanced immunoturbidimetric assay principle. Ferritin was measured on Cobas 6000 (e601 module) Roche (Germany) by applying electrochemiluminescence immunoassay (ECLIA) specifically the sandwich principle. Ferritin results were determined by 2-point calibration and a master curve provided via the reagent barcode.

## Biomarkers and definitions used to assess ferritin level

IDA was evaluated based on hemoglobin, serum ferritin, and CRP concentration. Hemoglobin concentrations were corrected for altitudes by subtracting 0.8g/dl per 500m altitude to get the estimated sea level value (*Sullivan et al., 2008*; *World Health Organization, 2011a*). Anemia was defined as hemoglobin concentration adjusted for altitude less than 11g/dl and classified as normal (≥11.0 g/dl), mild (10.0–10.9 g/dl), moderate (7.0 –9.9 g/dl), and severe (<7.0 g/dl) (*Sullivan et al., 2008*).

Ferritin concentration was adjusted for inflammation using correction methods recommended by WHO in 2020 (*World Health Organization, 2020*). First, the prevalence of low ferritin (<12 µg/L) was calculated on the basis of unadjusted ferritin estimates. Subsequently, the following three adjustment approaches were applied to account for inflammation under this study:

- Exclusion approach: The exclusion approach excluded individuals with elevated CRP concentration >5 mg/L, and calculated the prevalence of low ferritin in the remaining individuals considering ferritin concentration cut-off <12 µg/L.

- Higher ferritin cut-off: The higher ferritin cut-off adjustment approach uses a higher ferritin concentration cut-off <30 µg/L. There were two different approaches when the

high ferritin cut-off was used. The cut-off applied for the entire sample as well as the subset of individuals with elevated CRP as defined by a CRP concentration >5 mg/L. For this study, we used cut-off for the subset of individuals with elevated CRP (*World Health Organization, 2020*).

- Internal correction factor: The correction factor (CF) was calculated as the ratio of geometric means of the reference group (non-elevated CRP) to that of the respective inflammation groups (elevated CRP). We found a CF value equal to 0.85 (*Thurnham et al., 2010*; *Namaste et al., 2017b*; *Namaste et al., 2017a*). Then, to get the adjusted ferritin the subgroup with CRP>5mg was multiplied by 0.85 (*Gibson, 2005*; *World Health Organization, 2011b*). Subsequently, low ferritin was defined <12 µg/L for both groups of elevated CRP or not elevated CRP.

Finally, we used the higher ferritin cut-off adjustment approach because we got the higher prevalence of iron deficiency anaemia by this approach. Furthermore, we measured only CRP not have the data on AGP concentration and malaria, this could prevent us to rely completely on correction factor and the regression correction approach (*World Health Organization, 2020*). Iron deficiency anemia was defined as children having a combination of hemoglobin level of less than 11g/dl and adjusted low ferritin concentration (<12 µg/L) (*World Health Organization, 2014*; *World Health Organization, 2011b*).

## Anthropometric measurements

The weight and height of children were measured to assess nutritional status. Height was taken using a Seca213 height board (Seca 213; Seca GmbH, Hamburg, Germany) with a sliding headpiece while the child stood straight. Weight was taken using a calibrated Seca 874 electronic flat scale (Seca 874; Seca GmbH, Hamburg, Germany) with the child barefoot and wearing light clothing. Anthropometric measurements were calculated according to the Emergency Nutrition Assessment for SMART software 2011 (Toronto, Canada), developed using WHO child growth standards (*Juergen, Golden & Seaman, 2015*). Weight and height measurements were converted to height-for-age (HAZ), weight-for-age (WAZ), and weight-for-height (WHZ) z-scores, based on WHO reference standards. For WHZ, a z-score <-2 indicated wasting, ≥−2 to <2 indicated normal, and >2 indicated overweight. For HAZ, a z-scores<-2 indicated stunting, and >-2 indicated normal. WAZ z-scores<-2 indicated underweight and >-2 indicated normal. Moderate and severe undernutrition were defined as z-scores <-2 and <-3, respectively (*WHO Multicentre Growth Reference Study Group, 2006*).

## Dietary assessment

A 24-h dietary recall questionnaire and 7 days structured food-frequency questionnaires adapted to the local context from Food and Agriculture Organization (FAO) guidelines were used to collect information about dietary practices (*Food & Agricultural Organization, 2010*). The 24-h and 7 days child food frequency recall were computed in to nine food groups: cereals, roots and tubers; vitamin A-rich fruits and vegetables; other

fruit; other vegetables; legumes and nuts; meat, poultry and fish; fats and oils; dairy; and eggs. Dietary diversity score calculated by summing the number of food groups based on FAO guidelines, with scores ranging from low (≤3 food items), medium (4–5 food items), and high (≥6 food items).

The 24 h household dietary diversity was computed into twelve food groups: cereals, white tubers and roots, vegetables, fruits, meat, eggs, fish and other seafood, legumes, nuts and seeds, milk and milk products, oils and fats, sweets spices, condiments, and beverages (*Food & Agricultural Organization, 2010*). The Dietary Diversity score was based on the computation of the different food groups consumed and the volume (quantity) of food and nutrient intake were not measured.

The consumption of iron rich food was defined as if a child consumed at least one iron-rich food item among the following food groups: organ meat, flesh meat, or fish at any time in the last 24 h or at least once in the last 7 days preceding the interview.

The household food insecurity status was determined using the 9-component Household Food Insecurity Access Scale validated in Ethiopia (*Gebreyesus et al., 2015*). The HFIAS is composed of nine items, which are asked with a recall period of 1 month. For each item, there was a follow-up of the frequency of the occurrence question. Responses scored 'never' received a score of 0, 'rarely' scored 1, 'sometimes' scored 2, and 'often' scored 3, so that when summed, the lowest possible score was 0 and the highest 27. Household food security was categorized into four levels (food secure, mildly food insecure, moderately food insecure, and severely food insecure) based on the guidelines (*Coates, Anne & Paula, 2007*).

Furthermore, data on child morbidity (diarrhoea and cough for the last 15 days), child immunization, history of hospital admission, iron-fortified food consumption of children, and iron supplementation for the mother and child were collected.

## Socioeconomic status

The principal component analysis was used to assess the household wealth index. The presence or absence of each household item was coded as '0' for No and '1' for Yes. Variables included electricity access, as well as ownership of a radio, telephone, television, refrigerator, electric stove, bicycle, motorcycle, car, or computer. Ownership of land, domestic animals, household monthly income, and the number of people in the household, parent education, and occupation were excluded from the model. The first component explained most of the variance in the observed set of variables, and the final model explained 32% of the variance. We then created a wealth index using tertiles that were distributed into three groups: poor, medium, and richest (*Beaumont, 2012*).

## Statistical analysis

Data were double-entered and checked using EpiData v.3.1 (EpiData.dk, Odense, Denmark) and transferred to IBM SPSS v.20 (SPSS Inc., Chicago, IL, USA) for analysis. Descriptive statistics were used to summarize categorical variables. Means with 95% confidence intervals (CI), medians, and interquartile range were used to present

continuous variables. The bi-variable analysis was done to check the association between each independent variable with the IDA and iron-rich food consumption. All variables with $P$-value < 0.3 were included in the multivariable logistic regression analysis to retain some potential confounding variables. Variables with a $P$-value less than 0.05 were considered statistically significant. Factors associated with consumption of iron-rich food at least once in the last 7 days were determined. Multicollinearity among independent variables was checked and the variables with tolerance test <0.2 and variance inflation factor (VIF) >5.0 were excluded from the models.

The conceptual modelling approach published by Victora et.al was adopted to develop for this specific study which used to determine factors associated with iron deficiency anemia (Victora et al., 1997). In the multivariable analysis factors associated with IDA were determined using three-level regression analyses (level one underlying factors, level two intermediate factors, and level three immediate factors). Figure two described variables in the conceptual modelling developed for this study (Fig. S2).

## Data quality and control

To assure the quality of the data generated, standard operating procedures were followed during blood sample collection and all laboratory procedures. Structured questionnaires were translated into the local Sidamo and Amharic languages. A 2-day intensive training was given for data collectors and supervisors regarding study objective, questionnaire, interview techniques, anthropometric measurements, blood sample collection, and ethical issues during data collection. A pre-test was done among 17 mother-child-pairs in the nearest village outside of the study area before the actual study. The data collection was checked daily for accuracy, consistency, and completeness by the supervisor. Anthropometry and hemoglobin instruments were calibrated as per standards.

## RESULTS

### Child characteristics

From the sampled 340 children, 331 participated in the survey (97.4% response rate). Of the 331 children, 172 (52%) were girls, and the mean age was 39.2 (95% CI [38.1–40.6]) months. From the total of 331 children, only 10 (3%) children had received iron supplementation, and two of them were diagnosed with anemia and one was diagnosed with iron deficiency anemia. The children's illness history indicated that in the last 2 weeks preceding the survey, 100 (30%) children had watery diarrhea, and 71 (21%) had a cough. The nutritional status of children indicated that 125 (37%) children were stunted, four (1.3%) were wasted, and 72 (21%) were underweight (Table 1).

### Socio-demographic characteristics of mothers and households

A total of 331 mothers participated in the survey and their mean age was 27.7 years (95% CI [27.1–28.3]). The majority of mothers 231 (70%) were women working at home, 75 (22%) had no formal education, and 207 (62%) households were food insecure (Table 2).

**Table 1 Demography, morbidity, and nutritional status of children aged 24–59 months in Cheffe Cote Jebessa kebele, southern Ethiopia, 2017.**

| Variables | Characteristics | Frequency N = 331 | Percent |
|---|---|---|---|
| **Child sex** | | | |
| | Male | 159 | 48.0 |
| | Female | 172 | 52.0 |
| **Child age in months** | | | |
| | 24–36 | 145 | 43.8 |
| | 37–59 | 186 | 56.2 |
| **Mother response immunization completed** | | | |
| | No | 25 | 7.6 |
| | Yes | 306 | 92.4 |
| **Immunization completed confirmed in child health card** | | | |
| | No | 28 | 8.5 |
| | Yes | 207 | 62.5 |
| | No card | 96 | 29.0 |
| **Cough lasting 15 days** | | | |
| | No | 260 | 78.5 |
| | Yes | 71 | 21.5 |
| **Diarrhoea lasting 15 days** | | | |
| | No | 231 | 69.8 |
| | Yes | 100 | 30.2 |
| **History of hospital admission** | | | |
| | No | 289 | 87.3 |
| | Yes | 41 | 12.4 |
| | Missing | 1 | 0.3 |
| **Iron-rich food within last 24 h** | | | |
| | No | 281 | 84.9 |
| | Yes | 50 | 15.1 |
| **Iron supplementation** | | | |
| | No | 321 | 97.0 |
| | Yes | 10 | 3.0 |
| **Iron-fortified food** | | | |
| | No | 316 | 95.5 |
| | Yes | 15 | 4.5 |
| **Child dietary diversity score** | | | |
| | Low | 242 | 73.1 |
| | Medium | 65 | 19.6 |
| | High | 24 | 7.3 |
| **Current breast feeding** | | | |
| | No | 229 | 69.2 |

| Table 1 (continued) | | | |
|---|---|---|---|
| Variables | Characteristics | Frequency N = 331 | Percent |
| | Yes | 101 | 30.5 |
| | Missing | 1 | 0.3 |
| **Height-for-age score** | | | |
| | Stunting | 125 | 37.8 |
| | Normal | 206 | 62.2 |
| **Weight-for-height score** | | | |
| | Wasting | 4 | 1.2 |
| | Normal | 304 | 91.8 |
| | Overweight | 9 | 2.7 |
| | Missing | 14 | 4.2 |
| Weight for age score | | | |
| | Underweight | 72 | 21.8 |
| | Normal | 259 | 78.2 |

**Note:**
Missing value observed under the variable weight for height score was not the actual missing but the result flagged with WHO reference using ENASMART software.

## Iron rich food consumption

The 24-h child dietary recall indicated that the mean dietary diversity was 2.8 (95% CI [2.6–3.0]) and the median was 3 (IQR 2–4, range 0–8). Most of the children 242 (73%) scored low on dietary diversity (Table 1), only 50 (15%) children had consumed iron-rich food (meat, organ meat, and fish). The diet of the children contained mostly starchy staples 326 (98.6 %). The 7 days food frequency data showed that 104 (33%) children consumed iron-rich food at least once within the last 7 days (Table 3). Furthermore, household iron-rich food intake was similar to child iron intake of the last 24 h, it showed that only 51 (15%) of the household consumed iron-rich food (Fig. 1).

## Magnitude of anemia and inflammation

The mean hemoglobin concentration of the children was 11.6 g/dl (95% CI [11.5–11.7]), and the median was 11.8 g/dl (IQR: [10.9–12.5]). The prevalence of anemia was 107/331 (32.3%). Of the total number of anemic children, only 1 (0.9%) child had severe anemia, 35 (32.7 %) had moderate, and 71 (66.4 %) had mild anemia. The CRP measurements indicated that 23/107 (21 %) of anemic children had some sort of inflammation (Table 4).

## Iron deficiency anemia (IDA)

The estimated prevalence rates of IDA varied when using different adjustment methods. It was low in the unadjusted ferritin for inflammation group (18.7%) and, its prevalence ranged from 22% with the exclusion approach, 23% with correction factor approach and 25% with adjustment methods (Table 5).

**Table 2 Demographic and socio-economic characteristics of mothers and households, in Cheffe Cote Jebessa kebele, southern Ethiopia, 2017.**

| Variables | Characteristics | Frequency $N$ = 331 | Percent |
|---|---|---|---|
| **Mothers' age in years** | | | |
| | <25 | 135 | 40.8 |
| | 25–30 | 133 | 40.2 |
| | >30 | 63 | 19.0 |
| **Mother's education** | | | |
| | No formal education | 75 | 22.7 |
| | 1–8 school years | 180 | 54.4 |
| | >8 school years | 76 | 23.0 |
| **Father's education** | | | |
| | No formal education | 47 | 14.3 |
| | 1–8 school years | 168 | 51.2 |
| | >8 school years | 113 | 34.5 |
| **Mother's occupation** | | | |
| | Housewife | 231 | 69.8 |
| | Employed | 100 | 30.2 |
| **Number of people in household** | | | |
| | <5 people | 222 | 67.1 |
| | >5 people | 109 | 32.9 |
| **Income per month** | | | |
| | Low | 196 | 59.2 |
| | Medium | 117 | 35.3 |
| | High | 18 | 5.4 |
| **Land ownership** | | | |
| | No | 188 | 56.8 |
| | Yes | 143 | 43.2 |
| **Livestock ownership** | | | |
| | No | 188 | 56.8 |
| | Yes | 143 | 43.2 |
| **Wealth index** | | | |
| | Poor | 126 | 38.1 |
| | Medium | 82 | 24.8 |
| | Rich | 123 | 37.2 |
| **Household food insecurity** | | | |
| | Secure | 124 | 37.5 |
| | Mild | 32 | 9.7 |
| | Moderate | 72 | 21.8 |
| | Severe | 103 | 31.1 |

Orsango et al. (2021), *PeerJ*, DOI 10.7717/peerj.11649

**Table 3 Twenty-four hour's dietary recall and 7 days food frequency recall.**

| Food group | 24-h recall | Frequency (N = 331) | Percent | 1 week-frequency recall | Frequency (N = 331) | Percent |
|---|---|---|---|---|---|---|
| Starchy stable | No | 5 | 1.5 | No intake | 2 | 0.6 |
| | Yes | 326 | 98.5 | Once | 32 | 9.7 |
| | | | | 2–4 times | 108 | 32.6 |
| | | | | ≥5 times | 189 | 57.1 |
| Dark green vegetables | No | 215 | 65.0 | No intake | 101 | 30.5 |
| | Yes | 115 | 34.7 | Once | 49 | 14.8 |
| | | | | 2–4 times | 145 | 43.8 |
| | | | | ≥5 times | 36 | 10.9 |
| Vitamin A rich fruit and vegetables | No | 231 | 69.8 | No intake | 90 | 27.2 |
| | Yes | 100 | 30.2 | Once | 36 | 10.9 |
| | | | | 2–4 times | 120 | 36.3 |
| | | | | ≥5 times | 85 | 25.7 |
| Fruit and vegetables recode | No | 230 | 69.5 | No intake | 289 | 87.3 |
| | Yes | 101 | 30.5 | Once | 25 | 7.6 |
| | | | | 2–4 times | 14 | 4.2 |
| | | | | ≥5 times | 3 | 0.9 |
| Organ meat | No | 330 | 99.7 | No intake | 326 | 98.5 |
| | Yes | 1 | 0.3 | Once | 5 | 1.5 |
| Meat and fish | No | 281 | 84.9 | No intake | 227 | 68.6 |
| | Yes | 49 | 14.8 | Once | 62 | 18.7 |
| | | | | 2–4 per week | 37 | 11.2 |
| | | | | ≥5 times | 5 | 1.5 |
| Egg | No | 302 | 91.2 | No intake | 239 | 72.2 |
| | Yes | 29 | 8.8 | Once | 33 | 10 |
| | | | | 2–4 times | 54 | 16.3 |
| | | | | ≥5 times | 5 | 1.5 |
| Legume and nuts | No | 216 | 65.3 | No intake | 149 | 45 |
| | Yes | 115 | 34.7 | Once | 46 | 13.9 |
| | | | | 2–4 times | 99 | 29.9 |
| | | | | ≥5 times | 36 | 10.9 |
| Milk and milk products | No | 236 | 71.3 | No intake | 149 | 45 |
| | Yes | 95 | 28.7 | Once | 35 | 10.6 |
| | | | | 2–4 times | 94 | 28.4 |
| | | | | ≥5 times | 53 | 16 |

## Factors associated with iron deficiency anemia (IDA)

Using higher ferritin cut-off adjustment approach we estimated the Odds Ratios for IDA. The adjusted odds ratio showed that the prevalence of IDA significantly decreased as the height for age z-score increased (AOR 0.7; 0.5–0.9) (Table 6).

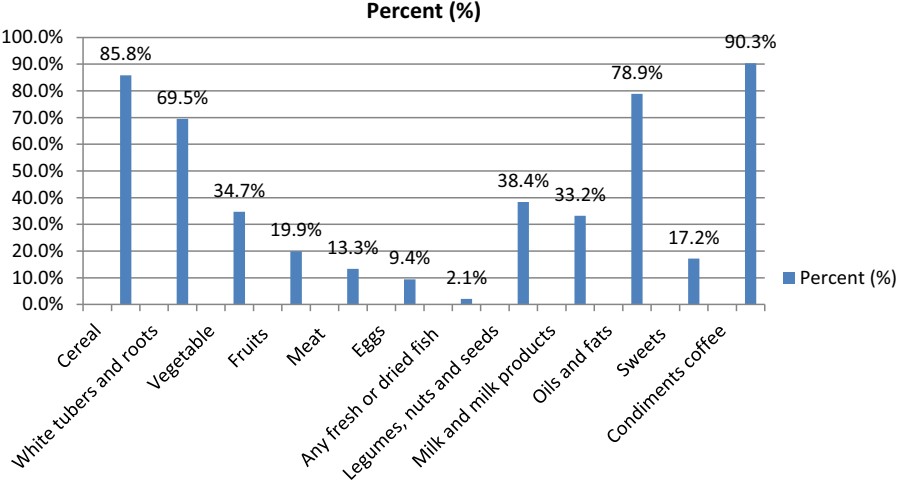

**Figure 1 Percentage of household dietary consumption in the last 24 h.**

**Table 4 Magnitude of anemia and inflammation among children aged 2–5 years in Cheffe Cote Jebessa kebele, southern Ethiopia.**

| Variables | Characteristics | Frequency | Percent |
|---|---|---|---|
| Anemia *N* = 107/331 | Severe | 1 | 0.9 |
| | Moderate | 35 | 32.7 |
| | Mild | 71 | 66.4 |
| Inflammation *N* = 107 | No (CRP < 5mg/L) | 84 | 78.5 |
| | Yes (CRP > 5mg/L) | 23 | 21.5 |

**Table 5 Median (IQR) of ferritin level and prevalence of iron deficiency anemia given different adjustment methods.**

| Correction method N = 107 | Median (IQR) Ferritin (µg/L) | Iron deficiency anemia *n* (%) | |
|---|---|---|---|
| | | No | Yes |
| Ferritin not adjusted | 25.7 (25.7–59.6) | 87 (81.3) | 20 (18.7) |
| Exclusion approach | 25.2 (12.8–35.3) | 65 (77.4) | 19 (22.6) |
| Correction Factor approach | 24.5 (12.6–34.6) | 82 (76.6) | 25 (23.4) |
| Higher ferritin-cut-off adjustment | 25.7 (25.7–59.6) | 80 (74.8) | 27 (25.2) |

Note:
CI, confidence interval; IQR, inter quantile range; µg/L, Micrograms per Litre.

## Factor associated with iron-rich food consumption

Iron-rich food consumption at least one time in the last 7 days indicated that mother's education (AOR 1.1; 1.0-1.2) and high household dietary diversity (AOR 1.4; 1.2–1.6) were positively associated with child iron-rich food consumption (Table S3).

**Table 6 Factors associated with iron deficiency anemia among children age 2–5 years in Cheffe Cote Jebessa kebele, southern Ethiopia, 2017.**

| N = 107 | | Iron deficiency anemia | | Crude OR (95% CI) | Adjusted OR (95% CI) |
|---|---|---|---|---|---|
| | | No (%) | Yes (%) | | |
| **Inherent factors** | | | | | |
| Age (continuous) | | | | 1.02 [0.98–1.06] | 0.96 [0.91-1.03] |
| Child sex | Male | 29 (36.3) | 12 (44.4) | 1 | 1 |
| | Female | 51 (63.8) | 15 (55.6) | 1.07 [0.47–2.47] | 0.91 [0.32–2.57] |
| **Underlying factors** | | | | | |
| Land ownership | No | 55 (68.8) | 14 (51.9) | 1 | 1 |
| | Yes | 25 (31.3) | 13 (48.1) | 1.59 [0.69–3.67] | 1.91 [0.69–5.30] |
| **Intermediate factors** | | | | | |
| Household food security | Food insecure | 55 (68.8) | 18 (66.7) | 1 | 1 |
| | Food secured | 25 (31.3) | 9 (33.3) | 0.97 [0.41–2.33] | 0.76 [0.25–2.28] |
| Height for age (HAZ) (continuous) | | | | 0.86 [0.73–1.00] | **0.74 [0.56–0.98]** |
| Weight for height (WHZ) (continuous) | | | | 0.99 [0.63–1.56] | 1.22 [0.73–2.06] |
| **Immediate factors** | | | | | |
| Child dietary diversity | Low | 61 (76.3) | 18 (66.7) | 1 | 1 |
| | Medium and high | 19 (23.8) | 9 (33.3) | 1.60 [0.62–4.16] | 1.79 [0.57–5.58] |
| Meal frequency (continuous) | | | | 1.48 [0.81–2.70] | 1.80 [0.89–3.62] |
| Green vegetable consumption per week | 0 intake | 19 (23.8) | 7 (25.9) | 1 | 1 |
| | At least once | 61 (76.2) | 20 (74.1) | 0.63 [0.24–1.64] | 0.59 [0.19–1.83] |

**Note:**
Variables with a *P*-value < 0.3 were included in multi-variable analysis. Bold format *P* ≤ 0.05; CI, confidence interval; OR, odds ratio.

## DISCUSSION

This study has revealed that one-fourth of the anemia cases were due to IDA, and the 24 h food frequency recall showed that only 15% of children consumed iron-rich food and 30% of children consumed iron-rich food at least once in the last week preceding the survey. These findings are comparable with 20% of iron-rich food consumption by children less than 5 years of age within the last 24 h reported in Ethiopia (*Tiruneh et al., 2020*). Iron deficiency anemia was associated with a low height for age of the children, while low iron-rich food consumption was associated with mother's education and household dietary diversity score. In our study, the prevalence of anemia was 32%, which is comparable with the prevalence studies in northern Ethiopia for children in the 6–59 months age group (*Gebreegziabiher, Etana & Niggusie, 2014*).

The 25% prevalence of iron deficiency anemia in our study was higher than the Ethiopian National Micronutrient Survey which estimated a 9% prevalence of iron deficiency anemia among pre-school children in 2016. To our knowledge, there are no earlier community based studies done in Ethiopia on iron deficiency anemia among young children, however, a study from south-western Ethiopia in school children reported a higher prevalence of 37% of iron-deficiency (*Desalegn, Mossie & Gedefaw, 2014*). The discrepancy may be due to age differences and because the socio-economic and dietary practices could vary across the regions. Studies from Rwanda showed a similar result with

our findings in the age range 6–59 months (*Rutayisire, Nwaike & Marete, 2019*), whereas in Angola 46% of anemia cases were because of iron deficiency in children 3–36 months of age (*Fancony et al., 2020*). Furthermore, a recent review finding indicated that 25% of anemia in young children was because of iron deficiency (*Green et al., 2017*).

We used different inflammation adjustment methods under this study to estimate IDA and we found some variation of the prevalence among these different methods (*World Health Organization, 2020*; *Namaste et al., 2017b*; *Thurnham et al., 2010*). The variation of proportion of iron deficiency anemia with inflammation adjustment could be explained by the high burden of inflammation among children in the study area (*Engle-Stone et al., 2017*; *Merrill et al., 2017*). Furthermore, the study shows that inflammation is one of the causes of iron deficiency anemia (*Fancony et al., 2020*).

In this study improved height-for-age was inversely associated with IDA, and similar findings were reported in children less than 5 years of age from Pakistan (*Habib et al., 2016*). This could be because iron deficiency may lead to delayed growth and development in children. A study from Qatar showed that treatment of IDA with iron supplementation improved the growth of children under 2 years of age (*Soliman et al., 2009*).

In our study iron deficiency anemia was not associated with iron-rich food consumption, however iron-rich food consumption was associated with demographic and socio-economic characteristics of the households. Nevertheless, the lack of association with IDA must be interpreted with care as only 107 cases were involved, and the study may not have been sufficiently powered to detect all risk factors statistically. However, studies from Rwanda and Brazil have also show that socio-demographic characteristics of the household were not associated with iron deficiency anemia (*Nobre et al., 2017*; *Rutayisire, Nwaike & Marete, 2019*). In this study, mothers with better education ate more iron-rich foods, as it is supported by other studies in Ethiopia (*Tiruneh et al., 2020*; *Choi et al., 2011*).

## Strength and limitation of the study

One of the strengths of this study is that it represents one of the few studies in Ethiopia that has assessed iron deficiency anemia using a representative sample of a community. The study was done in a farming community, and was a part of a larger study that aimed to evaluate the efficacy of home processed amaranth grain containing bread in the treatment of anemia among two-to-five year-old children in southern Ethiopia (*Orsango et al., 2020*). Even if the area represents a semi-urban part of a town, many of the household members practice agriculture typical of many areas in southern Ethiopia. Furthermore, iron deficiency anemia was assessed based on ferritin concentration adjusted for inflammation. A limitation of the study was that we did not assess potential causes of other types of anemia that could be prevalent in the area. Our sample size calculation was based on the EDHS 2011 report since our proposal development and approval was done before the release of 2016 EDHS report. Also children aged 3–5 years of age were used for sample calculation as EDHS were not reporting on the age group 2–5 years of age. Moreover, it was not possible to know the cause-effect relationship of the factors as it was a cross-sectional study. We used higher ferritin cut-off adjustment approach. Using a higher ferritin–concentration cut-off value for individuals with infection/inflammation

<30 μg/L was preferred to give a conclusion in our study. Because we didn't have the data on AGP concentration and malaria, results have to rely completely on the arithmetic correction factor approach and the regression correction approach.

## CONCLUSION

This study found that one-third of the children were anemic, and one fourth of the anemic children had iron deficiency anemia among children 2–5years of age in a semi-urban area of southern Ethiopia. Furthermore, the consumption of iron rich food by children was low, and iron deficiency anemia was linked to low height for age measures. Based on the WHO classification, IDA was a moderate public health problem among our study participants. Thus, both short and long term interventions should be implemented to mitigate the adverse effect of iron deficiency anemia in the study area. Short-term solutions could include food supplementation and fortification strategies, and long-term intervention should focus on improving food diversification, nutritional education, and promoting women's education. Furthermore, emphases should be given to reduce high burden of inflammation in the study area. We recommend further studies on area specific iron deficiency anemia to include AGP and malaria status to see any variation in the prevalence of IDA and study on potential causes of other types of anemia that could be prevalent in the area.

### Funding

This work supported by the Norwegian Programme for Capacity Development in Higher Education and Research for Development (NORHED) and the South Ethiopia Network Universities in public health (SENUPH No. ETH-13/0025). The funders had no role in study design, data collection and analysis, decision to publish, or preparation of the manuscript.

### Grant Disclosures

The following grant information was disclosed by the authors:
NORHED.
SENUPH: ETH-13/0025.

### Competing Interests

The authors declare that they have no competing interests.

### Author Contributions

- Alemselam Zebdewos Orsango conceived and designed the experiments, performed the experiments, analyzed the data, prepared figures and/or tables, authored or reviewed drafts of the paper, and approved the final draft.
- Wossene Habtu analyzed the data, authored or reviewed drafts of the paper, and approved the final draft.

- Tadesse Lejisa analyzed the data, authored or reviewed drafts of the paper, and approved the final draft.
- Eskindir Loha conceived and designed the experiments, performed the experiments, analyzed the data, prepared figures and/or tables, authored or reviewed drafts of the paper, and approved the final draft.
- Bernt Lindtjørn conceived and designed the experiments, performed the experiments, analyzed the data, prepared figures and/or tables, authored or reviewed drafts of the paper, and approved the final draft.
- Ingunn Marie S. Engebretsen conceived and designed the experiments, performed the experiments, analyzed the data, prepared figures and/or tables, authored or reviewed drafts of the paper, and approved the final draft.

### Human Ethics

The following information was supplied relating to ethical approvals (i.e., approving body and any reference numbers):

The institution's ethical board of Hawassa University (IRB/098/08) and the and Regional Ethical Committee West Norway (No. 2016/2034) provided ethical approvals.

### Field Study Permissions

The following information was supplied relating to field study approvals (i.e., approving body and any reference numbers):

Hawassa University College of Medicine and Health Science (No. SPH/1471/09).

### Data Availability

The raw measurements are available in the Supplemental Files.

### Supplemental Information

Supplemental information for this article can be found online at http://dx.doi.org/10.7717/peerj.11649#supplemental-information.

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
