# Peer review of "Iron deficiency anemia among children aged 2–5 years in southern Ethiopia: a community-based cross-sectional study"

_PeerJ, doi:10.7717/peerj.11649_

## Round 0.1 · original submission · Major Revisions

Authors,

The reviewers have attended to your work. Major revisions are required.

Kindly address all concerns raised, provide relevant details, yet succinctly. The editor looks forward to your revised manuscript.

Reviewer 1 ·

Basic reporting

No comment

Experimental design

No comment

Validity of the findings

No comment

Additional comments

The research question, relevance , methodology, results/interpretation as well as conclusions derived are good.
There are lots of editorial issues. Authors should read through carefully to correct them. Some will be highlighted alongside some other comments.
1. Line 31 including replace with involving
2. Line 42 ‘last week’ replace preceding week
3. Line 89 focusing on the assessment of iron rich food
4. Line 106-107 to be rephrased for clarity
5. Line 131, ferritin was omitted.
6. Line 136-137. Rephrase sentence. .... transported in cold box to Ethiopia.
7. Line 138-139. Mention the principle of reaction of Cobas e601 and 501.
8. Line 143 ‘sea level’
9. Line 173 Anthropometry measurements
10. Line 311, comparable with prevalence
11. Line 312-13. Revise sentence. The prevalence of IDA......
12. Line 325-327. Revise this sentence; the Information being passed is not clear
13. Line 338. Rwanda and Brazil have also
14. Line 321-322. ...treatment of IDA with iron supplementation
15. Line 334-335. Are the associations between demographics /socioeconomic characteristics and iron rich food consumption/IDA? The sentence does not read as such. Rephrase for clarity.
16. Line 355. Furthermore not further.
17. Line 357. ......public health problems among our study participants
18. Line 461-463, repeated in line 464-466
19. According to the table 1, 10 children included in the study were on iron supplements. Were any of them in the IDA group? This information should be in the write up for readers’ information.

Reviewer 2 ·

Basic reporting

Generally OK
Abstract:
Reword sentence beginning with: “Three-5 mL of venous blood” to clarify this was only one tube….

Experimental design

Generally OK
Methods:
Was a traceable reference material used to calibrate the ferritin assays?

Validity of the findings

Results:
L. 247. Data cited for underweight do not seem to match numbers on Table 1.
L. 275-276 – Data cited for moderate and mild anemia do not seem to match Table 4.

Additional comments

Timely and useful paper given the 2020 WHO Guideline on Use of Ferritin. Presentation is straight forward and will be particularly valuable to those working in research and policy.

L. 164 and references– Gibson, RS is the correct citation (Rosalind is the given name)
Discussion:
Specifically, what was the reason for selection of the "Higher ferritin-cut-off adjustment" approach for reporting your IDA data?
More in-depth comparisons among the current results and additional papers from the BRINDA project might be useful. Consider [1-3]
Editorial Comments:
Line 54 and references – Central
Line 143 – sea
Review throughout for noun and verb plural/singular agreement.

1. Namaste, S.M., et al., Methodologic approach for the Biomarkers Reflecting Inflammation and Nutritional Determinants of Anemia (BRINDA) project. The American Journal Of Clinical Nutrition, 2017. 106(Suppl 1): p. 333S-347S.
2. Engle-Stone, R., et al., Predictors of anemia in preschool children: Biomarkers Reflecting Inflammation and Nutritional Determinants of Anemia (BRINDA) project. The American Journal Of Clinical Nutrition, 2017. 106(Suppl 1): p. 402S-415S.
3. Merrill, R.D., et al., Factors associated with inflammation in preschool children and women of reproductive age: Biomarkers Reflecting Inflammation and Nutritional Determinants of Anemia (BRINDA) project. The American Journal Of Clinical Nutrition, 2017. 106(Suppl 1): p. 348S-358S.

Reviewer 3 ·

Basic reporting

The opening sentence in line 53-54 is not indicated the most recent figure. please check the most recent figure from global nutrition report 2020.

You assumption (line 56) about the causes of anemia is not the same any more. Please read

This review

file:///C:/Users/bekel005/Downloads/nutrients-08-00693.pdf


Global experts opinion on harvest plus website

https://www.harvestplus.org/viewpoints/new-findings-iron-deficiency-anemia-experts-weigh

I recommend you to revise your manuscript based on the new findings from this review and experts view in the field.

There is repetition of sentence like line 56 and 62.

you second paragraph don't have all the figures required. you mentioned poor, low, most … It's better to indicate the actual figures to the readers.

Experimental design

You have been using and justifying the prevalence in the background from demographic and health survey 2016. Why you calculate your sample size using central statistics agency 2012 figure ?

when exactly you collect the data ?

your previous publication referred says from February 15, to March 30, 2017 ? but the sample size was 100. Can you tell us the duration of this study for 340 children?

Who did the blood sample collection

why the two machine for laboratory analysis ?

did you use internal quality control?

How do you know if the machine is reading the right quantity?

What method of laboratory analysis used?


We don’t know how you used the 24 hr recall data? What you did before data analysis?


Was the 24 hr dietary recall qualitative or quantitative or semi quantitative?

in line 198, we don't know which questionnaire you are referring. is it 24 hr dietary recall or 7 day FFQ?

For which dietary data you applied FAO nine food categories ? is it for 7 day recall or 24 hr dietary recall ?

line 202 - Now you added another dietary assessment method. Please descript each dietary method separate: how you collect the data? What you did before data analysis? what analysis you did at the end.

What you described in the dietary assessment section is mix of all methods and it’s not detail enough.

Validity of the findings

The conclusion is not clear to which community, and geographic area.

Any reason why all variables P-value< 0.3 included in the multiple logistics regression?

Your finding about child immunization card is not clear. What does this result mean? What if the mother lost immunization card which is most common?

current breast feeding total n is 330 where as your percentage is 100%. You mentioned in the title your N is 331?

Something is wrong with this figure and variable names. please check and report wasting and underweight separate with the right figure.

please check the Weight-for-height (wasting) score naming of sub categories (which says Under weight?)

Additional comments

Make sure you comparison your findings with other similar findings are not with different age categories and geographic coverage. If you did that please make it clear to your reader so that they will be aware of it. Your interpretation of findings has to be inline with the current information bout causes iron deficiency anemia.

Annotated reviews are not available for download in order to protect the identity of reviewers who chose to remain anonymous.

---

## Round 0.2 · Minor Revisions

Please, authors kindly attend to the comments raised by reviewers, in addition to checking the following:

1. Please ensure that prior to the objective statement, the rationale for this study is well explained.

2. In the statistical analysis, indicate clearly why the specific statistical approach is being used.

3. In the conclusions, indicate the limitations encountered from this study. In what ways would future studies help to address it?

Look forward to your revised manuscript

Reviewer 1 ·

Basic reporting

Fairly ok. Authors to pay attention to grammar use. In view of this make corrections to line 129-130 and 272.

Experimental design

Well defined

Validity of the findings

Ok

Additional comments

Generally, the revised article reads better as most queries has been revised.

Reviewer 2 ·

Basic reporting

No comment.

Experimental design

Lines 134-140. “serum” is mentioned several times on these lines, but L 130 says blood was collected by using a lithium heparin plasma separator test tube. Clarify if ferritin and CRP were analyzed in serum or in plasma

Validity of the findings

No comment.

Additional comments

Line Comment
37 ..child had both..
48 ..women’s
54 ..anemia are due..
95 Medicine and Health Sciences
115 33%
137 ..kept in..
202 ≤3
207 ..and the volume..
208 ..intake were not measured.
215 Responses scored ‘never’..
219 ..guidelines.
243 ..variables was checked..
257 pairs
266 ..one was..
272 ..mothers participated..
273-274 ..home, 75%..
362 ..results have to..
374 ..women’s education

---

## Round 0.3 · accepted · Accept

Thank you authors for revising the manuscript, and addressing all concerns raised by the reviewers. The work is now acceptable for publication. This is a very useful study. The authors have benefitted from the peer-review process, which has elevated the quality of this work. Thank you for selecting PeerJ as your journal of choice, and we look forward to your future scholarly submissions. Congratulations :)